# Evaluation of the Effect of X-Ray Therapy on Glioma Rat Model Using Chemical Exchange Saturation Transfer and Diffusion-Weighted Imaging

**DOI:** 10.3390/cancers17152578

**Published:** 2025-08-05

**Authors:** Kazuki Onishi, Koji Itagaki, Sachie Kusaka, Tensei Nakano, Junpei Ueda, Shigeyoshi Saito

**Affiliations:** 1Department of Medical Physics and Engineering, Division of Health Sciences, Graduate School of Medicine, The University of Osaka, Suita 560-0871, Osaka, Japan; kazuki1003kazuki@gmail.com (K.O.); beitou@kuhp.kyoto-u.ac.jp (K.I.); nten16083@gmail.com (T.N.); uedaj@sahs.med.osaka-u.ac.jp (J.U.); 2Division of Clinical Radiology Service, Kyoto University Hospital, 54 Kawaharacho, Shogoin, Sakyo-ku, Kyoto 606-8507, Kyoto, Japan; 3Division of Sustainable Energy and Environmental Engineering, Graduate School of Engineering, The University of Osaka, Yamadaoka 2-1, Suita 565-0871, Osaka, Japan; kusaka@see.eng.osaka-u.ac.jp; 4Department of Radiological Sciences, Morinomiya University of Medical Sciences, Nankokita 1-26-16, Osaka 559-8611, Osaka, Japan; 5Department of Advanced Medical Technologies, National Cardiovascular and Cerebral Research Center, Suita 564-8565, Osaka, Japan; 6Immunology Frontier Research Center (IFReC), The University of Osaka, Suita 565-0871, Osaka, Japan; 7Premium Research Institute for Human Metaverse Medicine (PRIMe), The University of Osaka, Suita 565-0871, Osaka, Japan

**Keywords:** glioblastoma, X-ray, 7T-MRI, chemical exchange saturation transfer imaging, diffusion-weighted imaging

## Abstract

Several preclinical studies have been conducted on the efficacy of high-field MRI in treating brain tumors. This study aims to determine the effects of X-ray therapy on a rat model of brain tumors using magnetic resonance imaging (MRI). We performed MRI before and after X-ray therapy to determine the treatment effects on brain metabolites and water molecule diffusion. The results showed that X-ray therapy reduced the creatine concentration in the tumor and inhibited the increase in water molecule diffusion. We revealed the impact of X-ray therapy on brain tumors. The results obtained in this study may be beneficial for predicting prognosis.

## 1. Introduction

Glioblastoma (GBM) is the most common malignant intracranial tumor and has a low survival rate [1,2,3,4,5,6]. GBMs are difficult to remove by surgery; hence, multidisciplinary treatment combining radiation therapy and chemotherapy is commonly used in addition to surgery. GBM is generally treated using a combination of radiation therapy, which consists of 10 Gy per week for 6 weeks (total, 60 Gy), and chemotherapy with temozolomide. Despite treatments that combine surgery and adjuvant radiochemotherapy, patients’ outcomes remain poor, with a 5-year overall survival rate of 5.4% [7]. Few therapies are effective for GBM owing to various factors, including tumor invasiveness, immunosuppressive microenvironment, adaptive resistance to treatment, and intertumoral heterogeneity [8,9]. Therefore, it is necessary to identify effective treatments for GBM.

Chemical exchange saturation transfer (CEST) imaging is a non-invasive magnetic resonance imaging (MRI) technique that indirectly measures low-concentration metabolites such as creatine, glucose, and glutamate [10]. CEST has been used to study strokes, tumors, and epilepsy. The body primarily comprises bulk water, from which MRI signals are acquired. In bulk water, certain solutes resonate at frequencies different from those of bulk water protons and undergo chemical exchanges with them. When a saturation pulse was selectively applied to the protons of these solutes, the signal from the solute protons became saturated. Over time, the bulk water signal decreased as unsaturated bulk water protons replaced saturated solute protons through chemical exchange. These solutes were observed indirectly by measuring the bulk water signal. Exchangeable protons include amino (-NH_2_), amide (-NH), and hydroxyl (-OH) [11,12,13,14]. CEST analysis utilizes the Z-spectrum and the magnetization transfer ratio (MTR) asymmetry, called the MTR curve. The Z spectrum was obtained by plotting the bulk water signals at the respective resonance frequencies of the biological substances. Various brain metabolite changes over time can be examined using MTR asymmetry. Previous studies have examined brain metabolite changes caused by X-ray irradiation of the GBM; however, few have utilized CEST imaging using 7T-MRI.

Hydrogen atoms in the body are constantly in irregular motion, which is referred to as Brownian motion. Diffusion-weighted imaging (DWI) can capture the Brownian motion and visualize minute movements. The apparent diffusion coefficient (ADC) quantifies the degree of motion and is beneficial for detecting acute stroke, abscesses, and tumors with high cell density (high malignancy) [15]. The ADC value also reflects the cell density, allowing the study of changes in cell density induced by X-ray irradiation [16]. Various studies have examined the effects of X-ray irradiation on water molecule diffusion in GBM using DWI. Studies combining CEST with DWI have also been conducted. However, few studies have analyzed the correlation between the MTR and ADC values on a pixel-by-pixel basis using CEST imaging and DWI of the same section and matrix size. This pixel-by-pixel analysis of MTR and ADC values may provide more precise imaging. This study aimed to examine the changes in brain metabolites and the diffusion of water molecules using CEST imaging and DWI after 15 Gy X-ray irradiation in a rat model of glioma.

## 2. Materials and Methods

### 2.1. Cell Culture

C6 rat glioma cells were purchased from the JCRB Cell Bank (National Institutes of Biomedical Innovation, Health, and Nutrition, Osaka, Japan). The cultures were incubated in Dulbecco’s modified Eagle’s medium supplemented with 10% fetal bovine serum and 1% penicillin-streptomycin solution at 37 °C and 5% CO_2_.

### 2.2. Animal Preparation

The Research Ethics Committee of the University of Osaka approved all the experimental protocols. All experimental procedures involving animals and their care were performed in accordance with Osaka University Guidelines for Animal Experimentation and the National Institutes of Health Guide for the Care and Use of Laboratory Animals. Animal experiments were performed using 7-week-old male Wistar rats (n = 12) purchased from Japan SLC (Hamamatsu, Japan). All rats were housed in a controlled vivarium environment (24 °C; 12:12 h light:dark cycle) and fed a standard pellet diet and water ad libitum.

Brain tumor rat models were created by implanting C6 rat glioma cells directly into the right brain. During the operation, rats were immobilized using brain retainers (Stereotaxic Instruments for Rats, NARISHIGESCIENTIFIC INSTRUMENT LAB, Tokyo, Japan) and anesthetized continuously using 2% isoflurane. An electric drill was used to create a hole 4 mm to the right of the bregma. Then, 4 × 10^5^/5 µL of C6 cells were injected into the striatum at a depth of 4 mm from the cranial surface at an injection rate of 1.0 µL/m over 5 min using a micro-syringe. X-ray irradiation (15 Gy) was performed on day 9 using an X-ray irradiation equipment (160 kV, 3 mA, 1 Gy/min; MediXtec, Chiba, Japan). The nose and body were shielded with lead, and the whole brain was irradiated. The study schedule is shown in Figure 1.

### 2.3. MRI

All experiments to acquire MR images of animal livers were performed using a horizontal 7-T scanner (PharmaScan 70/16 US; Bruker Biospin, Ettlingen, Germany) equipped with a transmit/receive volume radio frequency coil with a diameter of 40 mm. To obtain the MR images, the rats were positioned in a stereotaxic frame with a mouthpiece to prevent movement during acquisition. The rat’s body temperature was maintained at 36.5 °C with regulated water flow and continuously monitored using a physiological monitoring system (SA Instruments Inc., Stony Brook, NY, USA).

T2-weighted images (T_2_WIs) were acquired with the following parameters: repetition time (TR) = 3200 ms; echo time (TE) = 33 ms; Rapid Acquisition with Relaxation Enhancement (RARE) factor = 8; slice thickness = 1.0 mm; field of view = 32 × 32 mm^2^; matrix size = 256 × 256; number of slices = 10; number of averages = four; and scan time = 6 m 49 s. DWIs were acquired with the following parameters: TR = 2000 ms; TE = 26.5 ms; segments = 4; diffusion directors = 30; b values = 0, 1000, 2000 sec/mm^2^; slice thickness = 1.0 mm; field of view = 32 × 32 mm^2^; matrix size = 128 × 128; number of slice = 1; number of average = 1, and scan time = 8 m 40 s. CEST images were acquired with the following parameters: TR = 2200 ms; TE = 33 ms; RARE factor = 8; slice thickness = 1.0 mm; field of view = 32 × 32 mm^2^; matrix size = 128 × 128; number of slices = 1; number of averages = 1; magnetization transfer saturation pulse = 3.0 μT; and scan time = 19 m 57 s. Z-spectrum data were acquired from CEST images with varying saturation frequencies from −4.8 ppm to +4.8 ppm in 0.3 ppm steps. The obtained T_2_WIs enclosed the entire tumor, and we superimposed the area on the region of interest (ROI). The ADC values were calculated from DWIs, and the MTR values were calculated from CEST images. In addition, the creatine CEST signal was defined as an MTR asymmetry of +1.8 ppm, so we conducted pixel-by-pixel scatter plot analyses using MTR values at 1.8 ppm and ADC values.

### 2.4. Histological Studies

Hematoxylin and eosin (HE) staining was performed for histological evaluation. After completing an MRI scan on day 17 after implantation, the rats’ brains were harvested and fixed in 10% paraformaldehyde. We asked the Sapporo General Pathology Laboratory to perform HE staining on four animals: two in the control group and two in the irradiated group. HE staining was observed under a fluorescence microscope (BZ-X810; KEYENCE CORPORATION, Osaka, Japan).

### 2.5. Statistical Analyses

The data are presented as mean ± standard deviation. Differences were compared using one-way analysis of variance followed by Tukey’s post hoc test. All analyses were performed using the Prism 8 software (GraphPad Software, San Diego, CA, USA). *p* < 0.05 was considered statistically significant (* *p* < 0.05, ** *p* < 0.01, and *** *p* < 0.001).

## 3. Results

### 3.1. Tumor Size

We calculated the tumor volume from T_2_WIs by enclosing the tumor in the ROIs. The tumor volumes in each group are shown in Figure 2. This study examined whether 15 Gy X-ray irradiation could suppress tumor growth in a rat model of glioma. The tumor volumes in the control group on days 8, 10, and 17 were 26.3 ± 20.8 mm^3^, 51.3 ± 36.2 mm^3^, and 152.2 ± 55.0 mm^3^, respectively. Conversely, the tumor volumes in the irradiated group on days 8, 10, and 17 were 18.6 ± 6.9 mm^3^, 23.0 ± 9.0 mm^3^, and 27.3 ± 7.3 mm^3^, respectively. The tumor volume in the control group increased significantly from days 10 to 17 (*p* < 0.01). In addition, the tumor volume in the irradiated group on day 17 was significantly lower than that in the control group on day 17 (*p* < 0.001). The tumor volume in the control group increased over time, whereas that in the irradiated group remained nearly unchanged.

### 3.2. ADC Value

T_2_WIs and ADC images for each group are shown in Figure 3A–L. This study examined the effect of 15 Gy X-ray irradiation on the diffusion of water molecules in a rat model of glioma. The ADC value of the tumor area in the control group increased over time. The ADC values of the tumor area on days 8 and 10 after implantation were low in both the control and irradiated groups. The ADC images of the control group on day 17 after implantation revealed higher ADC values at the tumor center and lower ADC values at the tumor margin. Conversely, the ADC value of the tumor area in the irradiated group remained nearly unchanged over time. The ADC images of the irradiated group on day 17 after implantation (approximately 1 week after irradiation) indicated that the ADC values of the tumor area were lower than those of the control group on day 17.

### 3.3. MTR Value at 1.8 ppm

T_2_WIs and CEST at 1.8 ppm images for each group are shown in Figure 4A–L. This study examined the effect of 15 Gy X-ray irradiation on creatine levels in a rat model of glioma. The MTR of the tumor area in the control group increased over time. The MTR values of the tumor area on day 8 after implantation were low in both control and irradiated groups. The MTR of the tumor area on day 10 after implantation was higher in the control group and remained lower in the irradiated group. The MTR at 1.8 ppm image of the control group on day 17 after implantation revealed lower MTR values at the tumor center and higher MTR values at the tumor margin. Conversely, the MTR of the tumor area in the irradiated group remained nearly unchanged over time. The MTR at 1.8 ppm image of the irradiated group on day 17 after implantation (approximately one week after irradiation) indicated that the MTR values of the tumor area were lower than those of the control group on day 17.

### 3.4. MTR Curve

We calculated the MTR values in the tumor on the CEST images every 0.3 ppm and created MTR curves. The MTR curves for each group are shown in Figure 5A–C. This study examined the effect of 15 Gy X-ray irradiation on brain metabolites in a rat model of glioma. On day 8 after implantation, the MTR values for each frequency in the control and irradiated groups did not change significantly. The MTR values at 1.5 ppm, 1.8 ppm, 2.1 ppm, and 2.4 ppm in the control group on day 10 were 4.9 ± 0.8%, 5.9 ± 0.7%, 7.0 ± 1.4%, and 6.1 ± 0.9%, respectively. Conversely, the MTR values at 1.5 ppm, 1.8 ppm, 2.1 ppm, and 2.4 ppm in the irradiated group on day 10 were 3.4 ± 1.0%, 4.3 ± 0.6%, 5.2 ± 1.1%, and 4.4 ± 1.4%, respectively. The MTR values at 1.5 ppm (*p* < 0.05), 1.8 ppm (*p* < 0.01), 2.1 ppm (*p* < 0.05), and 2.4 ppm (*p* < 0.05) in the irradiated group on day 10 decreased significantly compared to the control group.

The MTR values at 1.2 ppm, 1.5 ppm, 1.8 ppm, 2.1 ppm, and 2.4 ppm in the control group on day 17 were 4.2 ± 0.6%, 5.3 ± 0.6%, 6.4 ± 0.4%, 6.6 ± 0.2%, and 6.0 ± 0.8%, respectively. Conversely, the MTR values at 1.2 ppm, 1.5 ppm, 1.8 ppm, 2.1 ppm, and 2.4 ppm in the irradiated group on day 17 were 3.0 ± 0.4%, 4.2 ± 0.5%, 4.9 ± 0.5%, 5.2 ± 0.4%, and 4.5 ± 0.6%, respectively. The MTR values at 1.2 ppm (*p* < 0.05), 1.5 ppm (*p* < 0.05), 1.8 ppm (*p* < 0.01), 2.1 ppm (*p* < 0.001), and 2.4 ppm (*p* < 0.05) in the irradiated group on Day 17 decreased significantly compared with those of the control group.

### 3.5. ADC and MTR Value at 1.8 ppm on a Pixel-by-Pixel Basis

The ADC values were calculated on a pixel-by-pixel basis for the tumors on the ADC images, and the MTR values were calculated on a pixel-by-pixel basis for CEST at 1.8 ppm images. The ADC and MTR values at 1.8 ppm on a pixel-by-pixel basis for each group are shown in Figure 6A and Figure 6B, respectively. The ADC values on a pixel-by-pixel basis in the control group on days 8, 10, and 17 were 0.6 ± 0.06 (×10^−3^ mm^2^/s), 0.7 ± 0.08 (×10^−3^ mm^2^/s), and 0.8 ± 0.07 (×10^−3^ mm^2^/s), respectively. Conversely, the ADC values on a pixel-by-pixel basis in the irradiated group on days 8, 10, and 17 were 0.6 ± 0.05 (×10^−3^ mm^2^/s), 0.6 ± 0.03 (×10^−3^ mm^2^/s), and 0.7 ± 0.03 (×10^−3^ mm^2^/s), respectively. The ADC value on a pixel-by-pixel basis in the irradiated group on day 17 was significantly lower than that in the control group (*p* < 0.05). The ADC value on a pixel-by-pixel basis in the control group increased over time, whereas that in the irradiated group remained nearly unchanged.

The MTR values at 1.8 ppm on a pixel-by-pixel basis in the control group on days 8, 10, and 17 were 5.6 ± 1.3%, 5.6 ± 0.6%, and 5.9 ± 0.5%, respectively. Conversely, the MTR values at 1.8 ppm on a pixel-by-pixel basis in the irradiated group on days 8, 10, and 17 were 6.4 ± 1.6%, 4.3 ± 0.6%, and 5.0 ± 0.3%, respectively. The MTR values at 1.8 ppm on a pixel-by-pixel basis in the irradiated group on days 10 (*p* < 0.01) and 17 (*p* < 0.01) decreased significantly compared with those in the control group. The MTR values at 1.8 ppm on a pixel-by-pixel basis in the control group remained nearly unchanged over time, whereas those in the irradiated group decreased.

### 3.6. Scatter Plot on a Pixel-by-Pixel Basis

This study examined the effects of 15 Gy X-ray irradiation using the standard deviation and correlation between ADC and MTR value at 1.8 ppm. Pixel-by-pixel scatter plots of the ADC and MTR values at 1.8 ppm in the control group are shown in Figure 7A–D, and pixel-by-pixel scatter plots in the irradiated group are shown in Figure 8A–D. The ADC values on a pixel-by-pixel basis in the control group on day 8, day 10, and day 17 were 0.6 ± 0.06 (×10^−3^ mm^2^/s), 0.7 ± 0.08 (×10^−3^ mm^2^/s), and 0.8 ± 0.17 (×10^−3^ mm^2^/s), respectively. The standard deviation reflects the pixel-by-pixel variation and not the individual-by-individual variation. Thus, the standard deviation of the ADC values on a pixel-by-pixel basis increased dramatically, unlike the standard deviation of the ADC values on an individual-by-individual basis. Conversely, the ADC values on a pixel-by-pixel basis in the irradiated group on days 8, 10, and 17 were 0.6 ± 0.05 (×10^−3^ mm^2^/s), 0.6 ± 0.06 (×10^−3^ mm^2^/s), and 0.7 ± 0.11 (×10^–3^ mm^2^/s), respectively. Thus, the variation in ADC values on a pixel-by-pixel basis was suppressed by X-ray irradiation at 15 Gy. In addition, the MTR values at 1.8 ppm on a pixel-by-pixel basis in the control group on days 8, 10, and 17 were 5.6 ± 1.9%, 5.6 ± 1.8%, and 5.9 ± 2.4%, respectively. Conversely, the MTR values at 1.8 ppm on a pixel-by-pixel basis in the irradiated group on days 8, 10, and 17 were 6.4 ± 2.0%, 4.3 ± 1.9%, and 5.0 ± 1.5%, respectively. The variation in the MTR values at 1.8 ppm on a pixel-by-pixel basis was suppressed by 15 Gy of X-ray irradiation. We also examined the correlation between the ADC and MTR values at 1.8 ppm in this experiment. There were negative correlations between ADC and MTR values at 1.8 ppm in the control group on days 8 and 10 after implantation. However, the correlation was positive on day 17 post-implantation. Conversely, there was a negative correlation between the ADC and MTR values at 1.8 ppm in the irradiated group on day 10 after implantation (one day after irradiation), and the negative correlation continued on day 17 after implantation (approximately 1 week after irradiation).

### 3.7. HE Stains

The results of HE staining on day 17 after implantation are shown in Figure 9A,B. There was a significant amount of cytoplasm without cell nuclei in the tumor center in the control group. Cell nuclei were abundant at the tumor margins, suggesting ongoing cell proliferation. Conversely, the tumor centers in the irradiated group had few cell nuclei and contained areas necrosed by irradiation. The tumor margins contained cell nuclei; however, tumor gaps were observed. These results indicate that 15 Gy of X-ray irradiation caused tumor center necrosis and inhibited tumor margin proliferation.

## 4. Discussion

This study investigated the effects of 15 Gy X-ray irradiation on a rat model of glioma using 7T-MRI. X-ray irradiation at 15 Gy suppressed tumor growth and water molecule diffusion. We found that the MTR values of 1.8 ppm decreased following irradiation.

This study examined the effect of 15 Gy X-ray irradiation on tumor volume. The tumor volume in the control group increased remarkably from days 8 to 17 after implantation. Conversely, tumor volume in the irradiated group remained nearly unchanged. Takahashi et al. implanted tumors in mice and performed X-ray irradiation [17]. A previous study reported that irradiation with 2 Gy over six consecutive days (total dose of 12 Gy) did not significantly change the tumor volume between the control and irradiated groups during the treatment period. These findings suggest that approximately 15 Gy of X-ray irradiation is necessary to suppress an increase in tumor volume. Stessin et al. also performed 10 Gy of X-ray irradiation in a rat tumor model and examined the changes in tumor volume [18]. They found that the increase in tumor volume could be suppressed by approximately half; however, irradiation alone could not completely stop tumor growth. These findings also suggest that approximately 15 Gy of X-ray irradiation is necessary to completely inhibit tumor growth, regardless of whether single or fractionated irradiation is used.

We also studied the effects of 15 Gy X-ray irradiation on ADC values. The ADC values in the control group increased from day 8 to day 17. Conversely, ADC values in the irradiated group showed little change over time. Several studies have reported that early-stage malignant tumors usually show higher signal intensity on DWI than normal fibroglandular tissue, resulting in lower ADC values [19,20,21]. The decreased ADC values in malignant tumors may be due to increased cellularity, larger nuclei with more abundant macromolecular proteins, and less extracellular space [19]. The ADC values increase as tumor growth continues, and because of edema obstruction [22]. Therefore, the increase in ADC values in the control group in this study is consistent with that reported in previous studies and may reflect an increase in the diffusion of water molecules due to edema and abscess development. Conversely, the ADC values in the irradiated group showed little change, suggesting that 15 Gy X-ray irradiation suppressed the growth of edema and abscesses. However, previous studies reported inconsistent results. Kauppinen et al. treated tumors with radiation therapy and chemotherapy [23,24]. They found an increase in the water diffusion capacity due to the expansion of the extracellular lumen by apoptosis and necrosis, which led to an increase in ADC values. They concluded that the increase in ADC values after treatment reflected an increase in cell death within the tumor tissue associated with the treatment and asserted that an increase in ADC values would indicate better effectiveness of the treatment. The reason for this difference may stem from the differences in the treatments. The previous study utilized radiation chemotherapy, whereas this study focused solely on X-ray irradiation. According to other studies, necrosis caused by radiation therapy can occur in malignant gliomas, with a frequency of 3–24% after local irradiation [25,26,27]. Its frequency has increased with the combination of radiation therapy and chemotherapy [26,28]. We conclude that the increase in ADC values in previous studies reflects higher therapeutic effects on brain tumors. Although tumor growth was suppressed, X-ray irradiation alone did not eliminate the tumors. Additionally, based on the results of HE staining in the irradiated group, we observed areas of brain necrosis caused by irradiation. However, regions of high cell density that appeared to be tumors were also identified at the tumor margin. The increase in ADC values observed in prior research may be attributed to the almost complete disappearance of the tumor owing to more effective radiation chemotherapy.

This study investigated the effect of 15 Gy X-ray irradiation on MTR values. The MTR values of the irradiated group decreased significantly compared with those of the control group at frequencies ranging from 1.5 ppm to 2.4 ppm on day 10. Notably, the *p*-value was less than 0.01 at a frequency of 1.8 ppm. Previous studies have shown that MTR values at frequencies of 1.8 ppm away from free water reflect signal values from creatine [29]. The significant decrease in MTR values at 1.8 ppm in the irradiated group on day 1 and approximately 1 week after irradiation indicated a reduction in creatine concentrations in the tumor area. This may be because 15 Gy X-ray irradiation halted tumor growth, and the tumor did not produce the creatine necessary for growth due to necrosis. In a previous study utilizing similar CEST imaging, MTR values decreased in necrotic areas [30,31]. This was consistent with the decreased MTR values observed in the tumor center of the control group. The ability to detect changes in MTR values as early as 1 day after irradiation suggests that MTR values at a frequency of 1.8 ppm, reflecting creatine, may be a beneficial indicator of changes in tumor energy metabolism.

We obtained CEST and DWI images with the same dimensions and matrix size to generate pixel-by-pixel scatter plots of MTR and ADC values at 1.8 ppm. According to the results, the standard deviation of ADC values increased in the control group over time. Whereas it did not increase in the irradiated group. This indicated that the control group had higher ADC values at the tumor center and lower ADC values at the tumor margin, resulting in greater variability in ADC values. In the irradiated group, the tumor was smaller, and edema growth was suppressed, resulting in less variation in ADC values. The conventional ADC and MTR values were calculated as the average values within an ROI. The MTR curve represents the MTR calculated as the average value. However, there is a concern that precise image diagnosis may be difficult since some of the ROIs contain a mixture of high and low values. Pixel-by-pixel analysis of ADC and MTR values may provide more precise image diagnoses. We also examined the correlation between the ADC and MTR values. There was a strong negative correlation between the ADC and MTR values in both the control and irradiated groups on day 10. Conversely, there was a positive correlation in the control group and a negative correlation in the irradiated group on day 17. Nakajo et al. also created scatter plots of ADC and MTR values for brain tumors and examined their correlations [32]. They found a positive correlation between ADC and MTR values in patients with severe disease; as the tumor grade increased, both ADC and MTR values also increased. Edema and necrotic areas were barely observed on day 10, and their severity was low. However, a positive correlation began to emerge in the control group because of the high severity on day 17. Despite the negative correlation on day 10, the correlation became positive as time passed, and the severity increased, which is consistent with the results of previous studies. In contrast, a negative correlation was observed between days 10 and 17 in the irradiated group. No positive correlation was found in the irradiated group on day 17, suggesting that 15 Gy X-ray irradiation suppressed the increase in tumor grade. Based on the above results, pixel-by-pixel analysis of ADC and MTR values may be beneficial for determining therapeutic effects. A negative correlation indicated a strong treatment effect, whereas a positive correlation indicated a weak treatment effect. It is possible to determine the treatment effect of irradiation from a new perspective using a scatter plot of the ADC and MTR values. We believe that the results of this study, especially the pixel analysis, have great potential to be beneficial in clinical practice.

This study has some limitations. First, a single dose of 15 Gy X-ray irradiation was used in this experiment. This choice was based on a previous study that observed morphological changes in tumors following a 15 Gy single irradiation [33]. Fractionated radiation is commonly used in clinical practice [34]. Therefore, it is necessary to examine the effects of fractionated irradiation in rat models of glioma. Additionally, we need to consider imaging in different cross-sections for CEST and DWI. We took CEST and DWI images in only one cross-section, and the brain metabolites and diffusion of water molecules may vary between the center and periphery of the tumor. It is desirable to analyze tumors in multiple cross-sections and three dimensions. Additionally, although we used conventional statistical analyses, there have been advanced machine learning approaches that have been used for MRI [35,36]. We will incorporate these new approaches to reduce the time needed for MRI. Second, although we found that 15 Gy of X-ray irradiation could stop tumor growth, it could not eliminate the tumor, indicating that treatment with X-ray irradiation alone may not be effective. Therefore, it is promising to investigate the efficacy of temozolomide, an anticancer drug, against brain tumors. Finally, we performed various imaging techniques using 7T-MRI on days 8, 10, and 17. We found that the tumor volume in the irradiated group remained nearly unchanged, even approximately 1 week after irradiation. However, there is a possibility that the tumor may begin to grow again; therefore, it is necessary to continue imaging after this period. Furthermore, whole-brain irradiation was conducted; however, we must consider an irradiation method that minimizes the impact on normal cells. Stereotactic irradiation will be used to ensure that the rats can survive for several months after irradiation.

## 5. Conclusions

This study investigated the effects of 15 Gy X-ray irradiation on a rat model of glioma using 7T-MRI. X-ray irradiation at 15 Gy suppressed tumor growth and water molecule diffusion. We found that the MTR values of 1.8 ppm decreased by irradiation. We determined the effect of irradiation on pixel-by-pixel scatter plots of ADC and MTR values. Based on these findings, we conclude that DWI and CEST are beneficial for clarifying the therapeutic effects of irradiation on brain tumors.

## Figures and Tables

**Figure 1 cancers-17-02578-f001:**
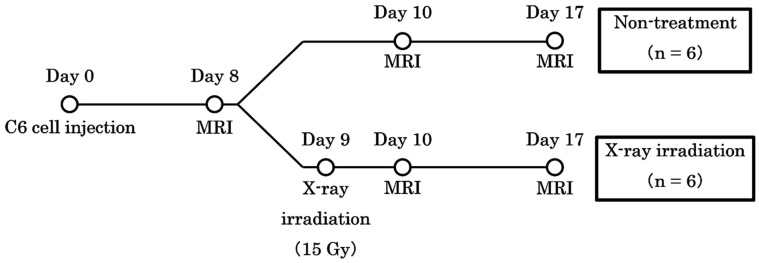
The schedule of the study. The date of implantation was set as day 0. Chemical exchange saturation transfer and diffusion-weighted imaging were performed on days 8, 10, and 17.

**Figure 2 cancers-17-02578-f002:**
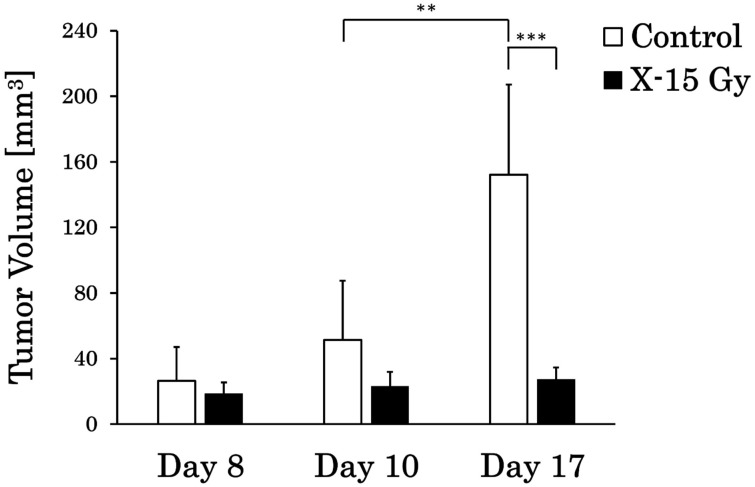
Tumor volume changes in the control and irradiated groups (** *p* < 0.01 and *** *p* < 0.001).

**Figure 3 cancers-17-02578-f003:**
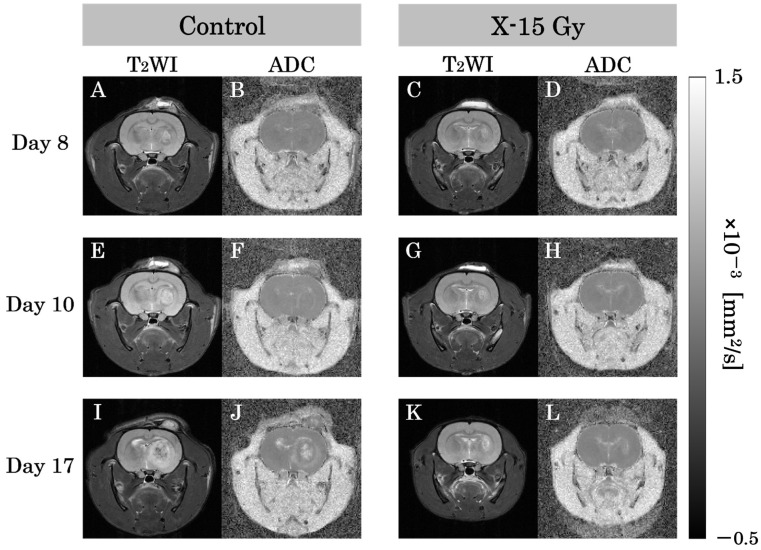
T_2_-weighted images (T_2_WIs) and apparent diffusion coefficient (ADC) images in each group are representative examples of rat heads. (**A**,**B**) show T2-weighted and ADC images of the control group on day 8 after implantation. (**C**,**D**) show images of the irradiated group. (**E**,**F**) show T2-WIs and ADC images of the control group on day 10 after implantation. (**G**,**H**) show images of the irradiated group one day after irradiation. (**I**,**J**) show T2-weighted and ADC images of the control group on day 17 after implantation. (**K**,**L**) show images of the irradiated group approximately 1 week after irradiation.

**Figure 4 cancers-17-02578-f004:**
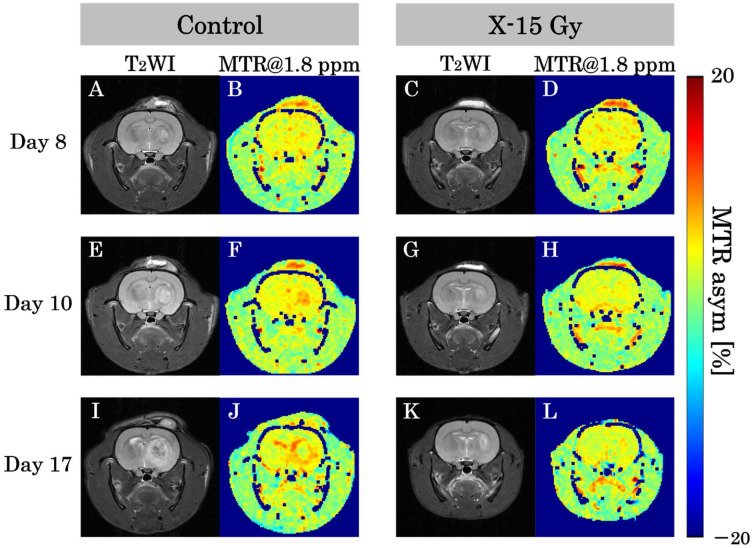
T_2_-weighted images (T_2_WIs) and magnetization transfer ratio (MTR) at 1.8 ppm images in each group show representative examples of the rat heads. (**A**,**B**) show T_2_-weighted and MTR at 1.8 ppm images of the control group on day 8 after implantation. (**C**,**D**) show images of the irradiated group. (**E**,**F**) show T_2_-weighted and MTR at 1.8 ppm images of the control group on day 10 after implantation. (**G**,**H**) show images of the irradiated group one day after irradiation. (**I**,**J**) show T_2_-weighted and MTR at 1.8 ppm images of the control group on day 17 after implantation. (**K**,**L**) show images of the irradiated group approximately 1 week after irradiation.

**Figure 5 cancers-17-02578-f005:**
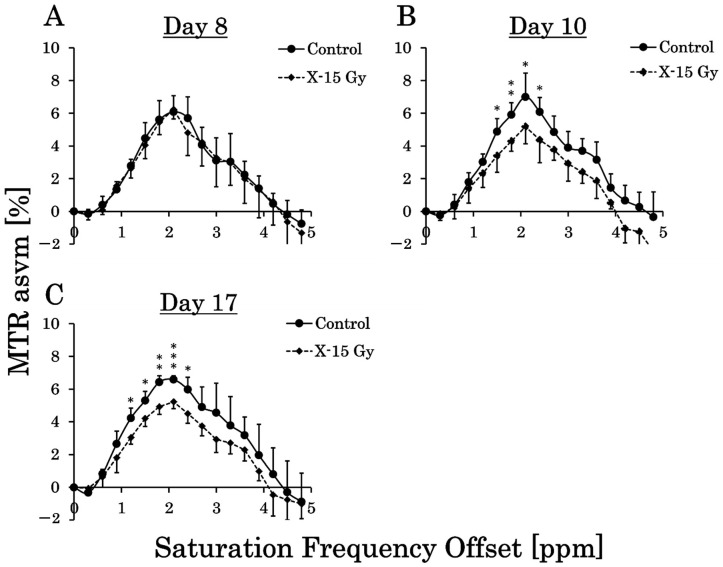
Magnetization transfer ratio (MTR) curves for each group show the average of the MTR values in each saturation frequency offset (* *p* < 0.05, ** *p* < 0.01, and *** *p* < 0.001). (**A**–**C**) are the MTR curves of the control and irradiated groups on days 8, 10 (one day after irradiation), and 17 after implantation (approximately 1 week after irradiation).

**Figure 6 cancers-17-02578-f006:**
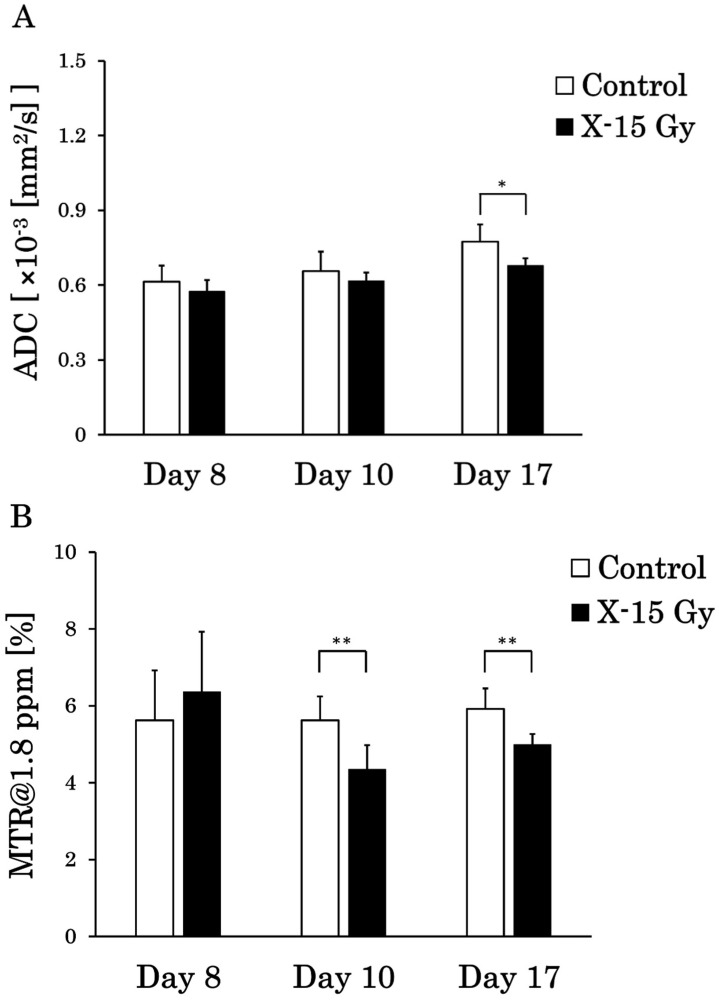
Apparent diffusion coefficient (ADC) and magnetization transfer ratio (MTR) values between the control and irradiated groups on a pixel-by-pixel basis (* *p* < 0.05 and ** *p* < 0.01). (**A**) is the ADC values on a pixel-by-pixel basis, and (**B**) is the MTR values at 1.8 ppm on a pixel-by-pixel basis in the tumor. The standard deviation is the individual-by-individual variation.

**Figure 7 cancers-17-02578-f007:**
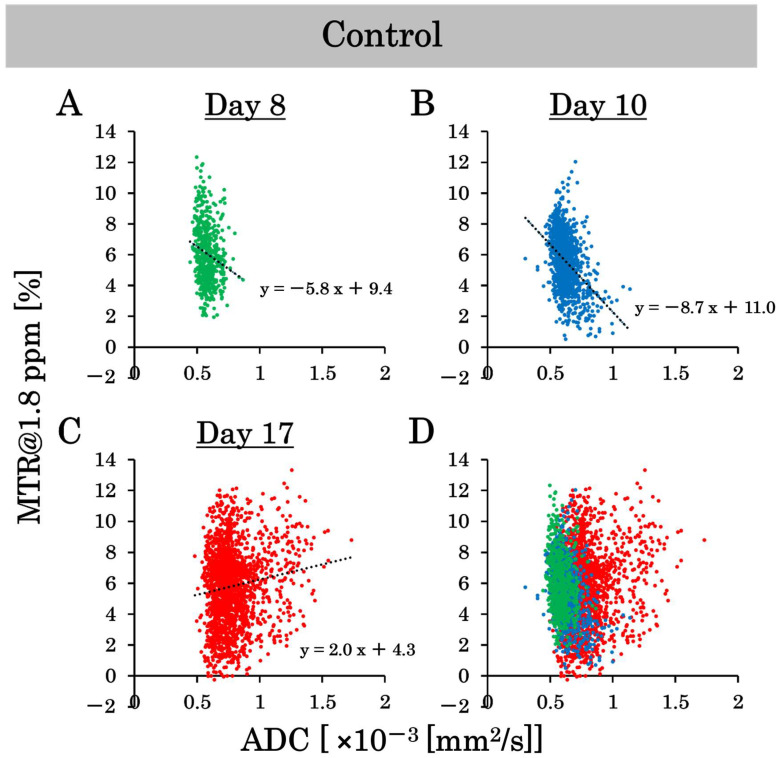
The scatter plots on a pixel-by-pixel basis of the apparent diffusion coefficient (ADC) and magnetization transfer ratio (MTR) values at 1.8 ppm show the variability in the control group. (**A**–**C**) are the scatter plots on a pixel-by-pixel basis on days 8, 10, and 17, and (**D**) is the combined scatter plot on a pixel-by-pixel basis in the control group on days 8, 10, and 17.

**Figure 8 cancers-17-02578-f008:**
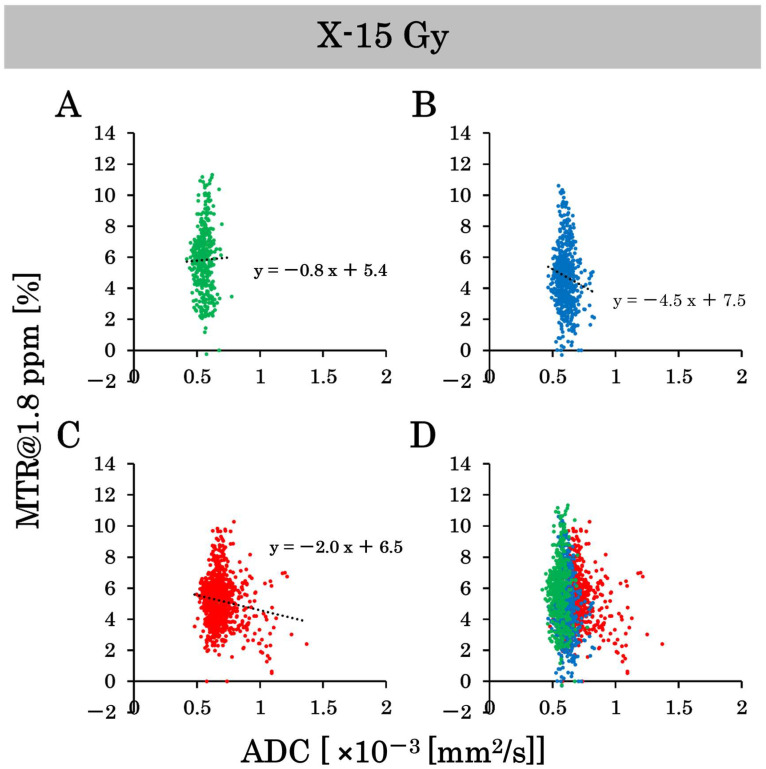
The scatter plots on a pixel-by-pixel basis of the apparent diffusion coefficient (ADC) and magnetization transfer ratio (MTR) values at 1.8 ppm show the variability in the irradiated group. (**A**–**C**) are the scatter plots on a pixel-by-pixel basis on days 8, 10, and 17. (**D**) is the combined scatter plot on a pixel-by-pixel basis in the irradiated group on days 8, 10, and 17.

**Figure 9 cancers-17-02578-f009:**
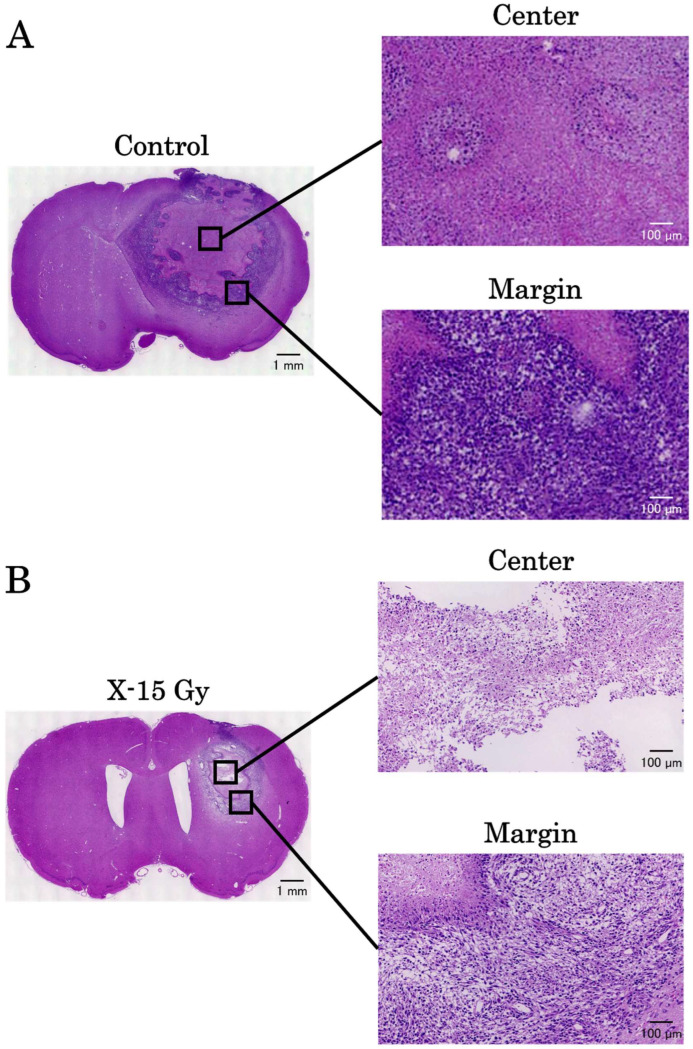
The results of hematoxylin and eosin stains on day 17 after implantation. (**A**,**B**) are the images of the control and irradiated groups, respectively.

## Data Availability

The data presented in this study are available on request from the corresponding author.

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
