# Peer review of "Evaluation of the Effect of X-Ray Therapy on Glioma Rat Model Using Chemical Exchange Saturation Transfer and Diffusion-Weighted Imaging"

_cancers, 2025, doi:10.3390/cancers17152578_

Round 1

Reviewer 1 Report

Comments and Suggestions for Authors
  1. How accurate and comparable is rat and human research model?
  2. How was the integration of glioma cells into brain tissue controlled and verified?
  3. It is hard to believe that 6 animals is a sufficient sample for statistical analysis and reliability.
  4. As mentioned above, the effects of fractionated radiation should be studied, which is what is used in clinical practice.
  5. What was the topicality of brain metabolite concentrations in the center and periphery of the brain, has this been investigated, and how might this be reflected in histological studies?

Author Response

Q1. How accurate and comparable is rat and human research model?

 Thank you for your valuable question. I will explain whether the results between rats and humans are comparable in terms of the experimental results of DWI. First, the ADC value, which is calculated from DWI images and indicates the degree of water molecule diffusion, is used to evaluate tumor malignancy in clinical practice. Some human clinical results have shown that the ADC value increases as the tumor grade increases. In this study, we used a rat model treated with 15 Gy of X-rays. As a result, ADC values were higher in the untreated rat model with higher malignancy compared to the treated group. In summary, the results of this study align with those observed in the clinical setting. Therefore, we believe that the results from this rat study can be compared with clinical results.

Q2. How was the integration of glioma cells into brain tissue controlled and verified?

 Thank you for your question regarding the tumor model. We have referred to previous studies on the transplantation method (Sachie Kusaka, Cells, 2024, 13, 1610). When transplanting tumor cells, it is common to do so into the striatum. If transplanted into the striatum, the tumor typically grows in concentric circles. If the transplant is unsuccessful in the striatum, the tumor cells tend to grow irregularly, making it difficult to control the tumor size. We performed an MRI eight days after implantation to confirm the tumor was correctly placed in the striatum. Individuals with tumors located far from the striatum were excluded from this study as cases of failed transplants. To accurately assess the morphological changes in tumors caused by the X-ray treatment, we ensured that the tumor sizes in the control and irradiated groups were nearly identical immediately before irradiation.

Q3. It is hard to believe that 6 animals is a sufficient sample for statistical analysis and reliability.

 Thank you for your precise comments on the statistical analysis. You are correct that the six-rat model may not be sufficient for statistical analysis due to the large individual variability. For example, we used 7T-MRI to perform CEST in this study, and the MTR values obtained from the CEST analysis varied widely among individuals. CEST imaging takes approximately one hour, during which time the rats' respiration must be kept constant. However, as days passed after implantation, tumors in the control group grew larger, and respiration became more irregular, increasing individual variability. On the other hand, since DWI imaging takes approximately ten minutes, it shows less variability compared to CEST. In addition, some rats with tumors grew too large and died during the experiment. It may be hard to observe rats for long periods and increase sample size with the current protocol. Therefore, we will consider a protocol that allows rats to survive until the end of the study to increase sample size in the future.

Q4. As mentioned above, the effects of fractionated radiation should be studied, which is what is used in clinical practice.

 Thank you for your valuable feedback regarding fractional irradiation. We are currently conducting similar experiments using this approach. Recently, we treated rats with 5 Gy of X-rays over three days (total 15 Gy). Results indicate that fractional irradiation may be more effective at tumor eradication than a single 15 Gy dose. We plan to publish detailed results of these experiments in the future. Although the total dose is not 15 Gy, a common clinical protocol is 2 Gy daily for five days. Therefore, we aim to align our research with clinical treatment methods. Additionally, combined therapy with temozolomide, an anti-cancer drug for brain tumors, is also employed clinically for more aggressive tumors. We intend to explore radiation combined with chemotherapy in the future studies.

Q5. What was the topicality of brain metabolite concentrations in the center and periphery of the brain, has this been investigated, and how might this be reflected in histological studies?

 Thank you for your important question about the topicality of brain metabolites. The CEST images reflecting creatine, one of the brain metabolites, are shown in Figure 4. In the control group, we can see that creatine is rarely found in the tumor center and is highly concentrated at the margins. Prior studies have also reported that as the malignancy increases, the necrosis expands in the tumor center, and creatine signal decreases. Thus, CEST can reveal the metabolite topicality within the brain tumors. These CEST findings are also consistent with HE staining results. In the control group, pronounced changes in creatine correlate with areas rich in nuclei and more proliferating cells in HE images. Additionally, CEST images show reduced creatine levels in the tumor center at 17 days post-transplantation. However, CEST images alone do not show histological differences in tumor centers between the control and irradiated groups. HE staining indicates that the control tumor center exhibits more edema areas, whereas the irradiated group shows more necrosis areas due to x-ray treatment. Combining CFST imaging with HF staining is crucial to comprehensively evaluate treatment effects. We believe that it is important to evaluate the effect of treatment using not only the CEST technique but also a combination of HE staining.

Reviewer 2 Report

Comments and Suggestions for Authors

The comments are reported in the uploaded file.

Comments on the Quality of English Language

The English language could be improved.

Author Response

 Thank you for your valuable feedback regarding future experiments. As you pointed out, we also believe that we need to monitor tumor changes beyond the 17th day after transplantation. In this study, we did not observe any rats with tumors growing again one week after irradiation. However, there is a possibility that tumors may grow back, so in future studies we will need to observe the rat model for as long as possible. We also focused on the correlation between ADC and MTR because few studies have evaluated treatment efficacy using these measurements. However, as you suggested, in the future we will examine these correlations along with their relation to histopathologic changes in HE staining. Your excellent feedback will help us develop a new research plan for the future.

Reviewer 3 Report

Comments and Suggestions for Authors

In this submission to Cancers, the authors examine the changes in brain metabolites and water molecule diffusion using chemical exchange saturation transfer (CEST) imaging and diffusion-weighted imaging (DWI). The authors implanted a glioma-derived cell line, C6, into the striatum of the right brain of 7-week-old male Wistar rats. CEST imaging and DWI were performed by the authors on days 8, 10, and 17 after implantation using a 7T-magnetic resonance imaging. The authors find that the irradiated group decreased significantly compared with those of the control group. The authors conclude that their study revealed the effects of 15 Gy X-ray irradiation in a rat model of glioma using CEST imaging and DWI.

I find this manuscript to be of interest to materials science researchers as well as readers of this journal. As such, I am generally supportive of publication with a few minor but essential comments. The authors use fairly conventional statistical analyses to analyze their results. However, there have been advanced machine learning approaches that have been used for MRI and biological research, which should be noted: Physica Medica 83, 2021, 79-87 and Environ. Sci. Technol. Lett. 2023, 10, 1017–1022. Specifically, these prior studies have shown that advanced machine learning approaches can provide more details into the underlying mechanisms of MRI and biological activity that can complement the experiments. I am not asking the authors to carry out new machine learning calculations at all, but these prior treatments should be noted since these machine learning techniques are well known and a mature field now.

The authors should also consider machine learning approaches in their manuscript to supplement the experimental findings.

Author Response

 Thank you for your valuable feedback on our machine learning approach. We used 7T-MRI to assess the effect of x-ray therapy on a rat model of brain tumor. As you mentioned, the MRI imaging process was quite time-consuming. Especially for CEST, it takes about one hour for imaging, and several hours for analysis. In this study, we analyzed the MRI data using MATLAB. We also used a standard one-way ANOVA for statistical analysis. The CEST technique is rarely used in clinical practice because of its disadvantages, such as the considerable time required and the lack of useful indices for brain metabolites. As we work toward bringing CEST technology to clinical use, we plan to incorporate machine learning approaches that will reduce the time needed for both CEST imaging and analysis.

Round 2

Reviewer 2 Report

Comments and Suggestions for Authors

The authors stated that findings may have potential implications for prognosis prediction. I suggest its publication as starting point of future research.

Reviewer 3 Report

Comments and Suggestions for Authors

The authors addressed my concerns; I support publication